# Fabrication of a Controlled-Release Core-Shell Floating Tablet of Ketamine Hydrochloride Using a 3D Printing Technique for Management of Refractory Depressions and Chronic Pain

**DOI:** 10.3390/polym16060746

**Published:** 2024-03-08

**Authors:** Tahmineh Karami, Emad Ghobadi, Mohammad Akrami, Ismaeil Haririan

**Affiliations:** 1School of Engineering, University of Tehran, Tehran P.O. Box 11155-4563, Iran; tahmine.karami@ut.ac.ir; 2Department of Pharmaceutics, Biomaterials, Medical Biomaterials Research Center, Faculty of Pharmacy, Tehran University of Medical Sciences, Tehran P.O. Box 14155-6451, Iran; emad.q47@gmail.com (E.G.); haririan@tums.ac.ir (I.H.); 3Department of Pharmaceutics, Faculty of Pharmacy, Tehran University of Medical Sciences, Tehran P.O. Box 14155-6451, Iran

**Keywords:** 3D printing, additive manufacturing, ketamine, tablet, extended release, personalized medicine

## Abstract

In this study, a novel floating, controlled-release and core-shell oral tablet of ketamine hydrochloride (HCl) was produced using a dual extrusion by 3D printing method. A mixture of Soluplus^®^ and Eudragit^®^ RS-PO was extruded by a hot-melt extrusion (HME) nozzle at 150–160 °C to fabricate the tablet shell, while a second nozzle known as a pressure-assisted syringe (PAS) extruded the etamine HCl in carboxymethyl cellulose gel at room temperature (25 °C) inside the shell. The resulting tablets were optimized based on the United States pharmacopeia standards (USP) for solid dosage forms. Moreover, the tablet was characterized using Fourier-transform infrared (FTIR) spectrum, scanning electron microscopy (SEM), differential scanning calorimetry (DSC), and buoyancy techniques. The results showed a desired dissolution profile for a 100% infill optimized tablet with total drug release (100%) during 12 h. Weight variation and content uniformity of the tablets achieved the USP requirements. SEM micrographs showed a smooth surface with acceptable layer diameters. According to the FTIR analysis, no interference was detected among peaks. Based on DSC analysis, the crystallinity of ketamine HCl did not change during melt extrusion. In conclusion, the floating controlled-release 3D-printed tablet of ketamine HCl can be a promising candidate for management of refractory depressions and chronic pain. Additionally, the additive manufacturing method enables the production of patient-tailored dosage with tunable-release kinetics for personalized medicine in point-of care setting.

## 1. Introduction

The conventional therapies for depression have been concentrated mainly on altering serotonin, norepinephrine, and dopamine neurotransmitter systems by medication of various antidepressant classes such as tricyclic antidepressants (TCAs), selective serotonin reuptake inhibitors (SSRIs), serotonin–norepinephrine reuptake inhibitors (SNRIs), and monoamine oxidase inhibitors (MAOIs). However, some treatment-resistant depressions are reported that their management needs alternative medication [1]. One of the medications for this purpose is ketamine HCl, an N-methyl-D-aspartate (NMDA) antagonist, mainly used as an anesthetic [2]. Ketamine HCl has been used in treating many chronic pain syndromes, including chronic neuropathic pains with various causes, complex regional pain syndrome, fibromyalgia, peripheral nerve injury, post-herpetic neuralgia, and spinal cord injury [3]. A study has established that IV infusion of sub-anesthetic doses of ketamine HCl can induce a rapid decline in depressive symptoms and suicidal ideations [4]. Further clinical studies have supported the effectiveness of ketamine HCl injections in treating depression and reducing suicidal behaviors [5,6,7]. The most commonly used reported infusion dose of ketamine HCl in depression treatment is 0.5 mg/kg over 40 min. However, the frequency of administration varies in different studies and the repeated dosing has been shown to prolong its efficacy [8]. It was reported that 4–6 infusions per two weeks extends the effectiveness period up to 20 days [9,10,11]. Studies with longer follow-up durations were warranted to assess the long-term efficacy of ketamine HCl treatment. In these studies, the safety and tolerability profile of ketamine HCl was acceptable and side effects associated with antagonism of NMDA and modulation of sympathetic and parasympathetic systems, such as dissociative symptoms, headache, nausea, drowsiness, dizziness, and changes in heart rate and blood pressure, have been reported. However, they were short-lasting, and none of them were severe enough to necessitate the termination of therapy [7,12].

In 2022, Fancy et. al carried out an investigation wherein intravenous administration of ketamine HCl was employed as a therapeutic intervention for individuals grappling with treatment-resistant bipolar I/II depression. This treatment regimen consisted of four sub-anesthetic doses of intravenous ketamine HCl, ranging from 0.5 to 0.75 mg/kg, administered over a duration of two weeks. The findings of the study indicated a notable alleviation of depressive symptoms across the entire study cohort, and there were progressively diminishing levels of anxiety and decreased incidence of suicidal thoughts following each successive ketamine HCl infusion [13].

The repeated IV administration of ketamine HCl is inconvenient due to the need for hospitalization, invasiveness of the procedure, high cost, and the emergence of the side effects mentioned above. Furthermore, some responses including tachycardia, hypertension, and dissociation have been reported to appear shortly after parenteral administration of ketamine HCl. However, the adverse effects have disappeared through oral administration, indicating that the antidepressant effects can be attributed to the active metabolite of ketamine HCl [14].

Interestingly, the evidence of greater tolerability, as well as minimized side effects, in treatment-resistant depression have been obtained for an extended release oral medication [15].

Therefore, alternative routes of administration, such as oral, have gained attention. The oral dose of ketamine HCl in various studies varies from 30 to 1000 mg per day, with frequency ranging up to 6 times daily [16]. Taking advantage of the oral form of ketamine HCl, sustainability in ketamine HCl delivery and necessity of weight-based dosing of ketamine HCl for chronic depression or resistant bipolar I/II depression treatment, a tunable formulation of controlled-release tablet proves advantageous. According to Hasan et al. (2021), oral administration of ketamine HCl has garnered significant attention due to the emergence of the secondary metabolite 2,6-hydroxynorketamine as a result of pre-systemic metabolism. Consequently, the administration of low doses of a recently developed extended-release ketamine formulation (PR-ketamine) has been observed to yield elevated plasma concentrations of the promising analgesic and antidepressant agents, namely 2R,6R-hydroxynorketamine and 2S,6S-hydroxynorketamine [17]. In 2023, Weiss examined the correlation between the in vitro dissolution rate and the in vivo absorption rate of extended-release ketamine HCl tablets to predict bioavailability through single-point sampling. In this regard, a floating system could be considered as a suitable option for producing extended-release tablets [18].

Floating tablet systems, characterized by their low density (less than 1.004 g/cm^3^), remain buoyant in the gastric environment without significantly affecting gastric emptying rates. This buoyancy enables the tablet to stay afloat in the stomach, facilitating drug release at an appropriate rate. Generally, such systems could be stayed in a float position in the stomach environment by one of two mechanisms, no effervescent systems or effervescent systems, which are distinguished by gas generation [19]. It is essential to note that the formulation strategy of the floating tablet must be tunable to address the personalized dosing requirements. The field of drug formulation development continually evolves to enhance existing drug formulations, discover novel excipients, and devise innovative methods to create tailored medicinal products. 3D printing technology, also known as additive manufacturing, represents the latest breakthrough in this domain. It offers advantages such as reduced prototyping time and costs, the ability to manufacture products of varying sizes, support for personalized medicine, and the creation of dosage forms previously deemed impossible [20]. In recent years, there has been a surge in pharmaceutical manufacturing via 3D printing, often referred to as “3D pharming” [21]. 3D printing in pharmaceuticals holds potential for incorporating multiple drug substances with varying doses within a single pill, managing drug release kinetics through formulation adjustments, and tailoring drug dosage to match the pharmacokinetic profile [22].

Fused deposition modelling (FDM) 3D printing is the most common method used in pharmaceutical applications due to the more significant number of biocompatible excipients, lower cost of necessary equipment, ability to create complex dosage forms, and easier production process [23]. Recently, producing filaments as well as 3D printing objects have been facilitated using the hot-melt extrusion (HME)-based 3D printing approach to fabricate pills with desired properties [22]. This method has allowed not only the mixing of diverse thermoset polymers in melt extrusion processes but also higher drug loading capacity, compared to the FDM-based 3D printing method.

Based on the literature review, this study introduces two significant advancements compared to previous research. The initial innovation involves the development of oral ketamine HCl tablets as an alternative to direct IV injection. This approach reduces hospitalization duration and mitigates the generation of the secondary metabolite 2,6-hydroxynorketamine due to pre-systemic metabolism. The second innovation is the utilization of 3D printing technology which enables the formulation of personalized and sustained-release ketamine HCl tablets according to the chronic or bipolar depression intensity and suicidal tendency. 

To achieve the mentioned innovation, in this study, for the first time, the HME/PAS-based 3D-printing method was applied to develop a tunable formulation, providing tailored doses of extended-release ketamine HCl tablets. In this regard, a floating core-shell tablet system was designed in order to remove the mentioned drawbacks besides improving adherence. Four different formulations of Soluplus^®^ and Eudragit^®^ RS-PO were tested for printing the shell of the tablet and a hydrogel-contained API was injected in pits of the shell. Finally, the best formulation was selected for the detailed characterization.

## 2. Materials and Methods

### 2.1. Materials

Ketamine hydrochloride solution (CAS: 1867-66-9) was purchased from Delphis pharmaceutical (Gujarat, Rajkot, India) and was freeze-dried to create a powder form with a purity of approximately 98.7%. Soluplus^®^ (CAS: 402932-23-4), Eudragit^®^ RS-PO (CAS: 33434-24-1), and carboxymethyl cellulose (CMC) (CAS: 9004-32-4) were provided from BASF (Ludwigshafenm, Germany), Degusa-huls Gruppe (Frankfurt, Germany), and Sigma-Aldrich (Darmstadt, Germany) companies, respectively. Ethanol solvent (CAS: 64-17-5, Purity ≥ 98%) was purchased from Razi manufacturing, Tehran, Iran.

### 2.2. Formulation of 3D Printing Tablet

Owing to the fact that the melting point of the ketamine HCl powder (~262 °C) was higher than the maximum melting ability of the 3D printing device (~210 °C), ketamine HCl was dissolved in hydrogel and the hydrogel itself was placed in the shell during the printing process. CMC was selected as the hydrogel polymer because it is soluble in gastric acid. An oval-shape tablet with a length, width, and height of 2, 0.7, and 0.5 cm was designed using Solidwork software (v. 2022) to deliver the desired amount of the drug. The interior compartment of the tablet contained four pits filled with 200 µL hydrogel polymer (20% *w*/*v*) containing 162.5 mg ketamine HCl suitable for adult patients. To achieve the best formulation in terms of extrudability, floating, and entire drug release within 12 h, different ratios of Soluplus^®^ to Eudragit^®^ RS-PO polymers were selected (Table 1) for printing the tablet shell. Accordingly, the best formulation was selected for further characterizations. 

### 2.3. 3D Printing Process

The aforementioned designed tablet was uploaded by the Repetier slicing software (v. 2022, Willich, Germany) connected to a 3D Bio-printer (BioFabX2, Omidafarinan Co., Tehran, Iran) equipped with a hot melt extruder having a 200 µM nozzle size and a pressure-assisted microsyringe (PAM) extruder.

According to the manufacturer, Soluplus^®^ glass transition temperatures (Tg) correspond to 70 °C and there is no thermal degradation under 220 °C. Tg of Eudragit^®^ RS-PO and thermal degradation are 64 °C and 170 °C, respectively [24]. The tablets with different formulations (Table 1) were printed by weighing 3 g of total polymer powder (Soluplus^®^ and Eudragit^®^ RS-PO), then mixing up by porcelain mortar and pestle and introducing to the HME cylinder of 3D printer apparatus.

The shell of the tablets was firstly printed under the temperature of 150–180 °C, depending on the specific formulation (F1 at 180 °C, F2 at 170–180 °C, F3 at 160–170 °C and F4 at 150–160 °C) at a speed of 18 mm/s. Following this, the second extruder (PAS), injected the ketamine HCl–CMC gel into the four pits of the tablet at ambient temperature (25 °C). Subsequently, the first extruder resumed printing to create the final four layers, sealing the top side of the tablet. The printer bed temperature was maintained at 50 °C throughout the process at a speed of 20 mm/s.

### 2.4. Characterization 

The morphology of the printed tablets was analyzed using SEM (TESCAN MIRA3, Brno, Czech Republic) at 20–25 kV with magnifications of 1000 µM, 200 µM, and 100 µM. To increase electron conductivity during the imaging, samples were covered by a highly thin layer of gold.

To obtain the FTIR spectrum of ketamine HCl 3D tablets, 5 mg of the tablet were powdered and mixed with 300 mg of potassium bromide, then 162.5 mg of the material were pressed into the matrix under 80 psi pressure. Finally, the spectrum was scanned in the range of 400–4000 cm^−1^ by PerkinElmer© instrument (Waltham, MA, USA).

The DSC analysis of the printed tablet was performed using a Universal V4.5A TA instrument (New Castle, DE, USA) to assess the molecular status of the tablet. The samples were analyzed in isolated aluminum pans from 50 °C to 300 °C in a thermal rate of 10 °C/min. Empty aluminum pans were used as the reference group. This analysis was performed under argon gas at a 40 mL/min rate.

The dissolution test of the 3D ketamine HCl tablets was carried out using 900 mL of a medium consisting of HCl (0.1 N) with USP apparatus 2 (paddle) operating at a rotation speed of 100 rpm for 12 h at 37 °C. Ketamine HCl release was analyzed using a UV-Vis spectrophotometer (Jasco UV-1500, Easton, PA, USA) at a wavelength of 263 nm. In order to eliminate the absorption interference of the polymers (Soluplus^®^/Eudragit^®^ RS-PO) and ketamine HCl, a tablet containing ketamine HCl and a tablet without ketamine HCl were dissolved in the dissolution medium (0.1 N HCl). After that, 1 ml of the liquid was sampled at 30 min, 1 h, 2 h, 3 h, 6 h, 9 h, and 12 h. The samples firstly were centrifuged at 10,000 rpm for 15 min to obtain a clear and transparent solution. Then, 500 µL of the supernatant was transferred to a 1 cm UV cell. The difference between absorbance of the tablet containing ketamine HCl and the tablet without ketamine HCl was considered as the main absorbance of ketamine HCl. To calculate the release percentages, a standard ketamine HCl solution with 0, 40, 80, 120, 160, and 200 µg in 0.1 N HCl solution was prepared and then the UV absorbance at 263 nm was measured to obtain the calibration curve (R^2^ = 0.9999) using a UV-Vis spectrophotometer (Jasco UV-1500, Hachioji, Japan). 

According to content of uniformity and weight variation in USP standards, the tablet was categorized as a single-unit container solid form with more than 25 mg of drug substance. Therefore, to perform content uniformity test, 162.5 mg ketamine HCl was dissolved into 900 mL of HCl solution (0.1 N) as standard ketamine HCl solution. Then, 10 tablets were dissolved separately in 900 mL of HCl solution (0.1 N). Absorbance of the standard solutions of ketamine HCl and test were recorded at a wavelength of 263 nm using a UV-Vis spectrophotometer (Jasco UV-1500, Easton, USA). Likewise, the absorbance interference of polymers as excipients with ketamine HCl was alleviated by the procedures described above and the calibration curve was performed as mentioned before. The ketamine HCl amount was determined compared to standards.

For weight variation test, 10 tablets were weighed and the relative standard deviations (RSD) were calculated [25].

According to the former studies of the authors on 3D printed tablets, the polymeric tablets have negligent friability according to the USP standards; therefore, the friability test was omitted [26]. 

Finally, the mechanical stress–strain test for 3D printed tablets was performed using a universal mechanical tester device (Zwick/Z100, Ulm, Germany). 

## 3. Results and Discussion

### 3.1. Tablet Formulation and 3D Printing Process

To fabricate a 3D-printed tablet by HME, melting API with polymers is a common method. However, in the present study it was not possible to melt API and polymers simultaneously because of the higher melting point of ketamine HCl powder (~262 °C), compared to the maximum melting capability of the 3D printing device (~210 °C). Therefore, we attempted to formulate a core-shell tablet of ketamine. Ketamine HCl was solubilized within a hydrogel, and this hydrogel was subsequently deposited into the shell during the printing process. The results of various formulations of the shell compartment are shown in Table 2. Soluplus^®^ is a graft copolymer containing PCL, PVA, and PEG components, demonstrating amphiphilic characteristics. Its compatibility with extrusion processes is attributed to its low glass transition temperature (Tg) [27].

Eudragit^®^ RS PO is a low porosity and low permeability polymer that is used for control-release tablets and has pH independent release. This polymer is used in different studies for making floating tablets [28]. Among the four formulations, F4 with an 80/20 (*W*/*W*) ratio of Soluplus^®^/Eudragit^®^ RS-PO revealed the best performance in terms of extrudability, printability, floating capability, and entire release within 12 h. Therefore, this formulation was selected for further characterization.

Ketamine, a BCS Class 1 drug, has shown not only high passive permeability but also no efflux by P-glycoprotein. Therefore, intestinal epithelium penetration for ketamine shows no rate limitation. 

The Soluplus^®^/Eudragit^®^ RS-PO mixture was extruded through hot-melt container by 3D printer to print the tablet shell at 150–160 °C. The CMC gel containing 162.5 mg ketamine HCl was subsequently injected inside the tablet pits at room temperature. The photos of the printed tablets are shown in Figure 1. It is worth noting that the Soluplus^®^ comprises a good plasticizer capacity and extrudability at the nozzle temperature; therefore, no plasticizer agent was added. The tablet design drawn by Solidwork software (v. 2022) and sliced design uploaded in the Repetier software (v. 2022) prior to the printing process are shown in Figure 2. The perspective view of four pits before injecting ketamine HCl-loaded CMC gel is obvious. 

### 3.2. SEM Analysis

The surface morphology of the printed tablet was analyzed by SEM technique at different scales (Figure 3). The results plainly showed smooth surfaces of the multiple deposited layers, which are laid perfectly beside each other. The merging of the layers is insignificant, and instances of merging or overlapping layers are due to disproportionate evaporation of polymer solvent. Considering the diameter of the HME nozzle equal to 200 µm, the micrographs show the mean diameter of 215 µm for parallel printed layers which is satisfactory.

### 3.3. FTIR Analysis

The vibration response of ketamine HCL using an infrared beam showed a peak at 3430 cm^−1^, indicating the N-H stretch of the amide group attached to the cyclohexanone. A small peak at 3060 cm^−1^ is due to the aromatic C-H stretching vibration. Moreover, the spectrum showed a peak at 2921 cm^−1^ which is attributed to the C-H vibration of the alkyl group in this frequency. The alkyl group is usually a non-aromatic CH_2_ or CH_3_. The peak at 1721 cm^−1^ is related to R_2_-C=O stretch, which is solid and typical of cyclic ketones stretch. In ketamine HCl, a carbonyl functional group is attached to the cyclohexanone ring. The band appearing at 1580 cm^−1^ is due to the C-N band vibration. The peaks between 1400 cm^−1^–1500 cm^−1^ result from the C-H bending, which is another mode of CH_2_ and CH_3_ vibrations for ketamine HCl. These peaks do not represent the C-H bands of aromatic carbons. The peaks at wavelength of approximately 1450 cm^−1^ are related to C-C stretching in the cyclohexanone ring (Figure 4a) [29].

According to Soluplus^®^ Spectrum, peaks appearing at 3457 cm^−1^, 2928 cm^−1^, and 1742/1645 cm^−1^ are attributed to –OH stretching, C-H stretching and C=O stretching in amide groups, respectively. For the polymer, CH_3_ bending appeared at 1443 cm^−1^.

The spectrum of excipients and the final tablet also displayed similar peaks to ketamine HCl, concluding that there was no covalent interaction between ketamine HCl, excipients, and the final tablet during the formulation and printing.

### 3.4. DSC Analysis

The DSC analysis diagram showing obtained heat flow (W/g) per applied temperature (°C) is illustrated in Figure 5. According to the diagram, no peaks were evident in the DSC thermograms of Soluplus^®^ and Eudragit^®^ RS-PO, signifying the amorphous state of these polymers, while a small peak appeared in the DSC thermogram of CMC at 150–170 °C. Denoting the melting point of ketamine HCl, DSC analysis of ketamine HCl showed an endothermic peak at approximately 250 °C., the DSC thermogram of the 3D-printed tablet displayed a minor endothermic peak at approximately 150 °C. This peak indicates the tablet melting point, encompassing both polymers and the active pharmaceutical ingredient (API). Importantly, this observed melting point matches the one observed during the 3D printing of the shell, which solely consists of polymers.

Notably, we observed an endothermic peak at 160 °C in the DSC thermogram of the final tablet, which differed from the 260 °C peak observed in the thermogram of pure ketamine HCl. This shift can be attributed to three main factors. Firstly, during the 3D printing process, the API was placed inside the hydrogel at room temperature. As the hydrogel was injected into the four pits of the tablet, there was a slight increase in the API temperature due to heat transfer between the shell polymers and the API. The comparison of results obtained from DSC analysis of ketamine HCl and the final tablet may demonstrate change in the crystallinity of ketamine HCl to amorphous state during the production process. However, XRD analysis is needed to confirm the precise status. Secondly, considering that the tablet mean weight is 1106 mg and the API weight is 165 mg, it is clear that the API is distributed within the polymers to a limited extent. Thirdly, we observed a small peak in the 150–170 °C range for CMC, which had an impact on the mentioned peak shift. In summary, the final tablet exhibited a minor peak at approximately 160 °C, rather than the expected 260 °C, due to these three factors.

### 3.5. Content Uniformity and Weight Variation

The results of the content of uniformity of the tablet are shown in Table 3. The amount of acceptance value of 5.73 is less than 15 which comply with USP standards.

To estimate weight variation of the tablets, ten 3D-printed tablets were weighted and their mean, standard deviation, and relative standard deviation (RSD) were calculated. The result showed an RSD percentage less than 5%, indicating the samples comply with USP standards (Table 4).

### 3.6. Dissolution Profile

The dissolution test was performed in accordance to the procedure described in Section 2.4 using 12 tablets. Upon contact with the dissolution media, the floating tablet undergoes a process wherein the polymer matrix swells, resulting in the formation of a gel-like structure that envelops the tablet. In this scenario, the release of the drug was accomplished through a dual mechanism: diffusion occurring through the gel-like structure and erosion of the polymer matrix [30].The results demonstrated the complete release of tablet contents within 12 h (Figure 6). At the beginning of the dissolution, the diffusion mechanism is significant compared with the swelling phenomenon, resulting in 28% release at the initial 3 h. However, the impact of polymers swelling (due to their interaction with the dissolution environment) decreased the diffusion influence, and the release reached 42% at 6 h (as indicated by a slope drop in the graph). Nevertheless, over time, as the erosion of the tablet shell increases, the effect of diffusion has once again escalated. At 9 and 12 h, the release has reached 81% and 100%, respectively. 

Formulating an oral ketamine HCl tablet with a steady drug release profile (which mostly last for up to 12 h), showed the advantages regarding the ability for out-patient treatment, lowering the side effects, and consequently increasing patient compliance. Such benefits could be achieved by 80/20 (wt.%) ratio of Soluplus^®^/Eudragit^®^ RS-PO, where F4 was the only formulation remained floating during the dissolution tests.

### 3.7. Buoyancy of the 3D-Printed Tablets

The results for the buoyancy of 3D-printed tablets were determined in HCl solution of 0.1 N due to the air trapped in hollow pits incorporated in the tablet and the use of low-density ingredients. Tablets maintained their buoyancy above the solution during a measured time of 12 h at 37 °C and 100 rpm (Figure 7).

According to the visual observations, swelling can be considered one of the release mechanisms and a major characteristic of the Eudragit RS-based 3D-printed tablet. Our previous experience in a study showed diffusion is another mechanism [31].

### 3.8. Mechanical Test of the 3D-Printed Tablet

The inclination of the initial linear segment of the stress–strain graph aligns with Young’s modulus (a gauge of material stiffness) [32]. The stress–strain chart for 3D-printed tablets is shown in Figure 8. It demonstrates that the tablets could withstand 190 newtons of stress, reaching their breaking point on strain. The decrease in strain around 90 N force can be attributed to the existence of pits inside the 3D-printed tablet. Therefore, two yield points appeared with 0.25 mm and 0.5 mm strain. However, the tablet stiffness has shown different results in the literature depending on the filament type, composition, and the API percentage [32].

## 4. Conclusions

In the current study, for the first time, we developed a tunable sustained-release oral tablet of ketamine HCl utilizing PAS/HME-based 3D printing for the purpose of controlling refractory depression and chronic pains. Among the formulations, the Soluplus^®^:Eudragit^®^ RS-PO ratio of 80:20 was selected as the best formulation for fabrication of ketamine HCl tables which was subjected to characterization techniques such as SEM, FTIR, and DSC. The results showed a 100% sustained release during 12 h, while neither the drug degradation nor polymer decomposition during the printing process was observed. Microscopy imaging confirmed satisfactory parallel printed layers of the shell. No interaction between ketamine HCl, excipients, and the final tablet was detected during the formulation and printing. The final tablet conforms to USP standards and has ulterior quality as well. Fabrication of the sustained-release ketamine HCl using the tunable additive manufacturing can be considered as a promising strategy to provide tailored dosage amounts for management of treatment-resistant major depression, as well as chronic pains.

According to the satisfactory results (drug release and characterization results) obtained in this in vitro study, the next objective of the authors would be to expand the investigation to the in vivo phase study.

## Figures and Tables

**Figure 1 polymers-16-00746-f001:**
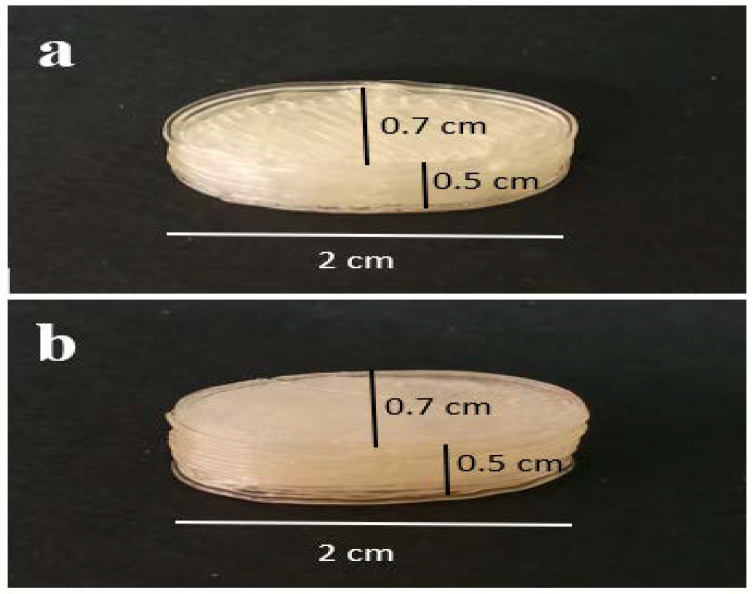
The photo of the 3D-printed tablet. (**a**) Top side and (**b**) bottom side view.

**Figure 2 polymers-16-00746-f002:**
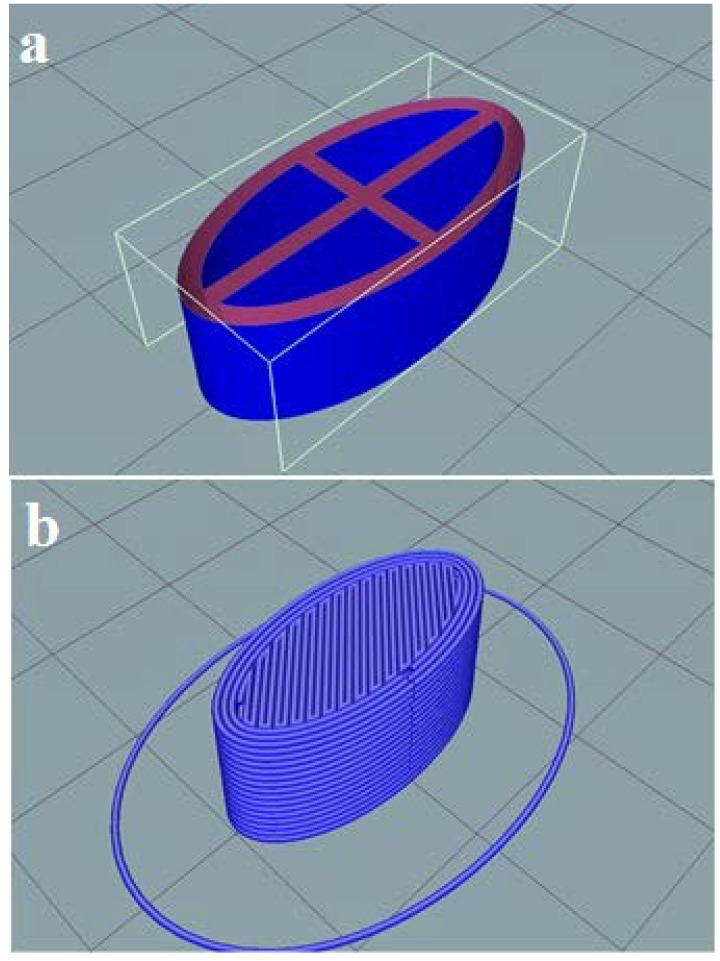
The shell design for ketamine HCl tablet using Solidwork (**a**) showing the perspective view of pits; (**b**) the perspective view of the sliced tablet design uploaded by Repetier software (v. 2022).

**Figure 3 polymers-16-00746-f003:**
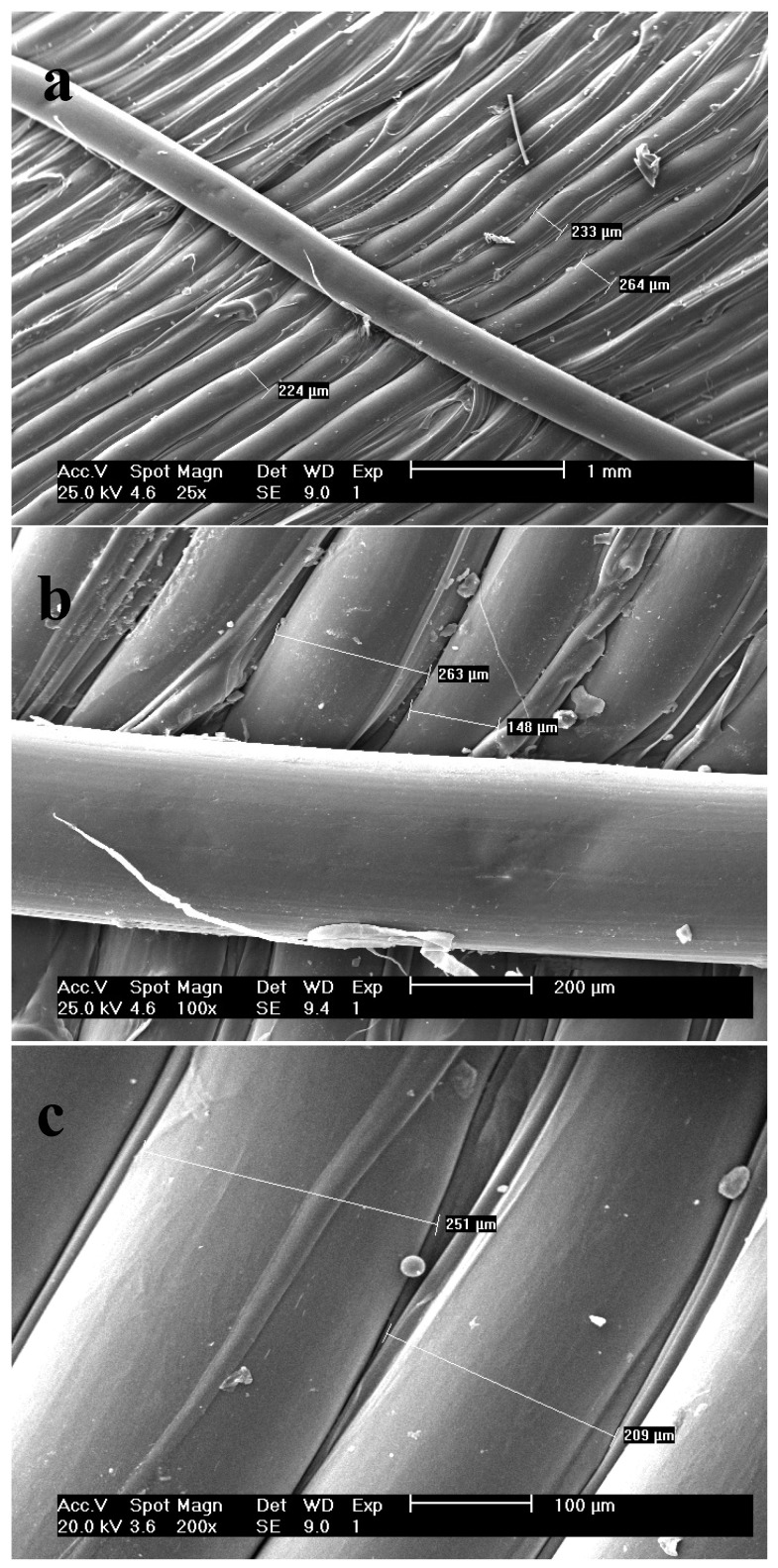
SEM micrographs of the surface of the tablet shell with scale of (**a**) 1000 µm; (**b**) 200 µm and (**c**) 100 µm.

**Figure 4 polymers-16-00746-f004:**
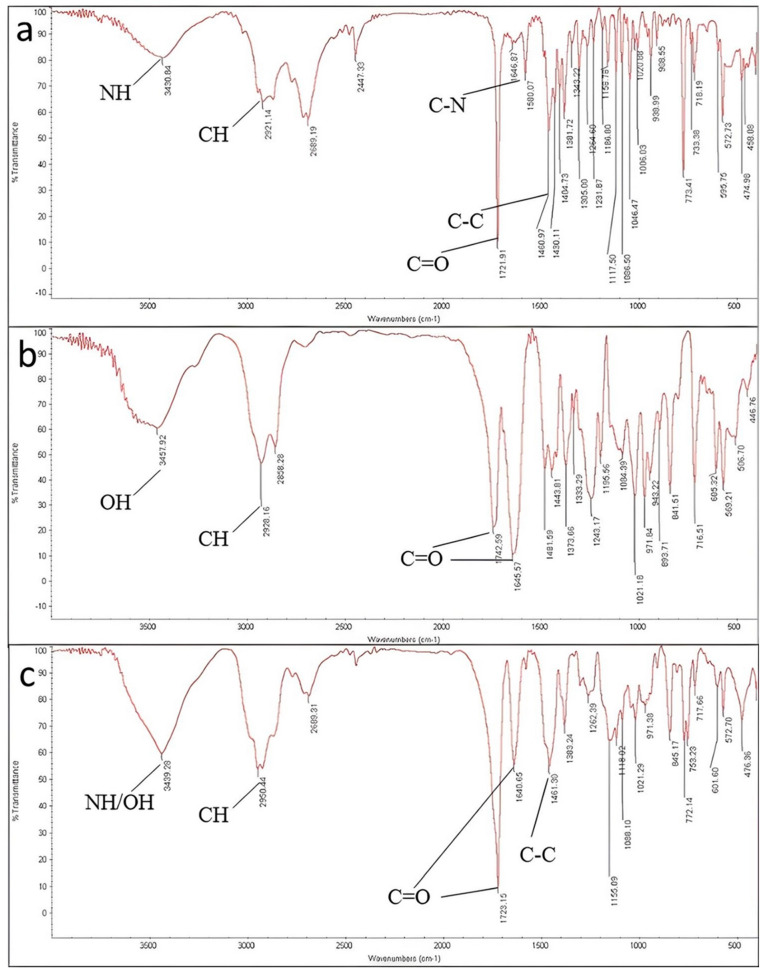
FTIR spectrums of (**a**) ketamine HCl; (**b**) Soluplus^®^ and (**c**) 3D-printed tablet in the range between 400 cm^−1^ and 4000 cm^−1^ using standard KBr method.

**Figure 5 polymers-16-00746-f005:**
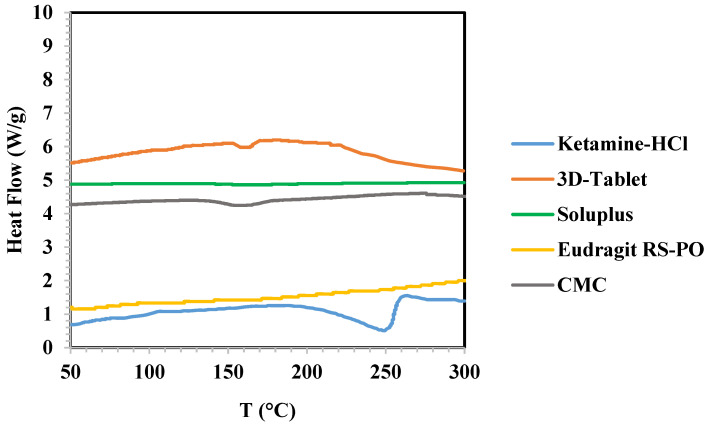
DSC analysis of ketamine HCl, Soluplus^®^, Eudragit^®^ RS-PO, and 3D-Tablet.

**Figure 6 polymers-16-00746-f006:**
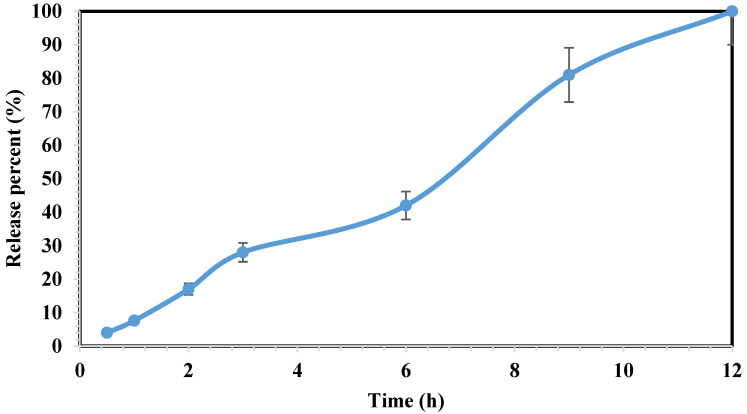
The dissolution profile of the 3D-printed tablets of ketamine HCl within 12 h. Drug release studied using UV spectrophotometry at 268 nm wavelength. Data represent mean ± SD, n = 3 for independent measurements.

**Figure 7 polymers-16-00746-f007:**
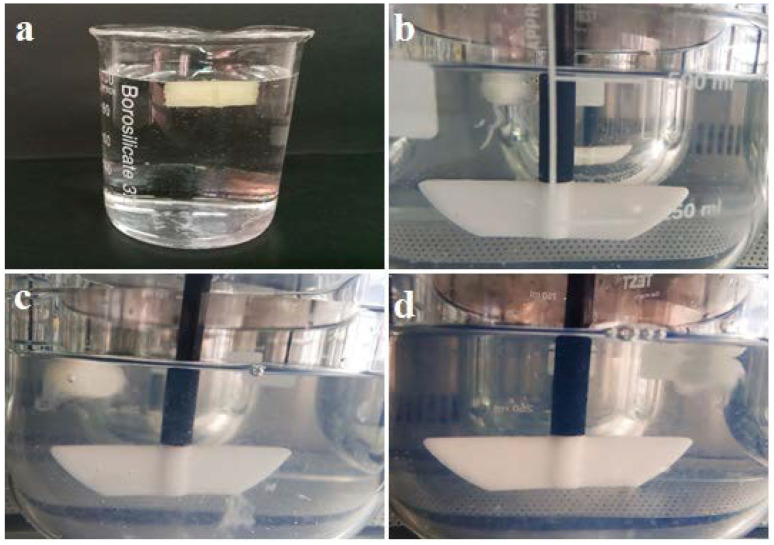
Demonstration of 3D-printed tablet in HCl solution of 0.1 N at 37 °C after (**a**) 1 h; (**b**) 3h; (**c**) 9 h and (**d**) 12 h.

**Figure 8 polymers-16-00746-f008:**
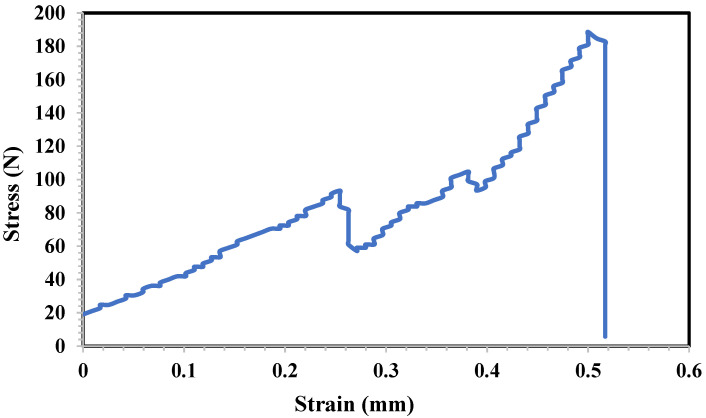
Mechanical stress–strain chart of 3D-printed tablet against mechanical force.

**Table 1 polymers-16-00746-t001:** Different ratio of Soluplus^®^: Eudragit^®^ RS-PO polymers for printing the tablet shell.

Formulation	F1	F2	F3	F4
Soluplus^®^	30%	50%	70%	80%
Eudragit^®^ RS-PO	70%	50%	30%	20%

**Table 2 polymers-16-00746-t002:** The results of various formulations of Soluplus^®^ and Eudragit^®^ RS-PO for printing the shell in terms of extrudability, floating capability and entire release within 12 h (* and ** marks indicate the favorable and unfavorable results, respectively).

Formulation	F1	F2	F3	F4
Soluplus^®^	30%	50%	70%	80%
Eudragit^®^ RS-PO	70%	50%	30%	20%
Extrudability	* ✓	✓	✓	✓
Floating	✓	✓	✓	✓
Complete release within 12 h	** ×	×	×	✓

**Table 3 polymers-16-00746-t003:** Determination of content of uniformity for 3D-printed ketamine HCl tablet (F4).

No.	Absorbance	Content Percentage
Tablet 1	0.3111	97.98
Tablet 2	0.3215	101.26
Tablet 3	0.3125	98.43
Tablet 4	0.3167	99.75
Tablet 5	0.3201	100.82
Tablet 6	0.3099	97.61
Tablet 7	0.3171	99.87
Tablet 8	0.319	100.47
Tablet 9	0.3087	97.23
Tablet 10	0.3126	98.46
Average (X)	0.31492	99.19
STD	0.0043	0.0135
RSD (S)	1.3636	1.3636
T = Target	100	
K for n = 10	4.20	
S	1.364	
Acceptance Value	5.73	

**Table 4 polymers-16-00746-t004:** Calculations of the weight variation of the 3D-printed ketamine HCl tablet.

Weight Variation										
No.	1	2	3	4	5	6	7	8	9	10
Weight (mg)	1097	1102	1172	1102	1001	1115	1155	1113	1101	1108
Mean	1106.6									
STD	42.49									
RSD	3.84									

## Data Availability

All data are included in the manuscript.

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
