# Peer review of "Fabrication of a Controlled-Release Core-Shell Floating Tablet of Ketamine Hydrochloride Using a 3D Printing Technique for Management of Refractory Depressions and Chronic Pain"

_polymers, 2024, doi:10.3390/polym16060746_

Round 1
Reviewer 1 Report
The main content of this manuscript describes a method for printing ketamine controlled-release oral tablets using the hot melt extrusion (HME) method. However, this manuscript is moderately innovative with less experimental supporting date. After review, I think the following problems need to be explained and solved.
1. The introduction to this manuscript does not capture the innovation well, so please add relevant references to highlight the problem addressed in this paper. It is recommended to consider starting from clinical needs or addressing clinical issues.
2. Providing more convincing data on drug release and specific methods for drug release determination
3. The diagrams in this manuscript are not standardized, such as figure 1 and figure 4. For example, FTIR requires specific labeling of the location of characteristic peaks.
4. The amorphous state of the Ketamine is not indicated by DSC alone, and it is recommended to supplement the XRD indication.
5. Supplement the DSC profile of the carboxymethylcellulose hydrogel and explain that none of the shell materials had a heat absorption peak, whereas the printed tablets had an absorption peak at 150℃.
6. The structure of the results and discussion sections of the article needs to be reconsidered and it is not recommended that an experiment be divided into one paragraph.
Based on the above comments, I personally believe that the paper is not suitable for publication in polymers at this time.
English needs to be improved.
Author Response
Detailed Response to Reviewer # 1
Thank you for your comments and suggestion concerning our manuscript. The comments and suggestions are all valuable and very helpful for revising and improving the manuscript, as well as the important guiding significance to our researches. We have studied comments carefully and have made correction to address the reviewer’s comments.
Comment of the reviewer:
- The introduction to this manuscript does not capture the innovation well, so please add relevant references to highlight the problem addressed in this paper. It is recommended to consider starting from clinical needs or addressing clinical issues.
Response of the authors:
Thanks for the great comment. The state of problem in the study has been mainly highlighted in lines 55-83 with the relevant references. In this regard, some case study results have been added. The clinical needs have been also mentioned in introduction.
“The key innovations associated in this paper was revised and added in lines 132-146:
Based on the literature review, this study introduces two significant advancements compared to previous research. The initial innovation involves the development of oral ketamine tablets as an alternative to direct IV injection. This approach reduces hospitalization duration and mitigates the generation of the secondary metabolite 2,6-hydroxynorketamine due to pre-systemic metabolism. The second innovation is the utilization of 3D printing technology which enables the formulation of personalized and sustained-release ketamine tablets according to the chronic or bipolar depression intensity and suicidal tendency.
To achieve the mentioned innovation, in the study, for the first time, the HME/PAS-based 3D printing method was applied to develop a tunable formulation, providing tailored doses of extended-release ketamine tablets. In this regard, a floating core-shell tablet system was designed in order to remove the mentioned drawbacks beside to improving adherence.”
Therefore, a new formulation consisting Soluplus: Eudragit RS-PO in shell compartment and carboxymethylcellulose gel containing drug in core compartment was developed.
The Core-shell word was added to the title, too.
Comment of the reviewer:
- Providing more convincing data on drug release and specific methods for drug release determination.
Response of the authors:
The dissolution test of the 3D Ketamine HCl tablets was carried out using 900 mL of a medium consisting of HCl (0.1N) with USP apparatus 2 (paddle) operating at a rotation speed of 100 rpm for 12 hours at 37°C. Ketamine HCl release was analyzed using a UV-Vis spectrophotometer at a wavelength of 263 nm. In order to eliminate the absorption interference of the polymers (Soluplus®/Eudragit® RS-PO) and ketamine HCl, a tablet containing ketamine HCl and a tablet without ketamine HCl were dissolved in the dissolution medium (0.1 N HCl). After that, 1 ml of the liquid was sampled at 30 min, 1h, 2h, 3h, 6h, 9h, 12h. The samples firstly were centrifuged at 10,000 rpm for 15 min to obtain a clear and transparent solution. Then, 500 µl of the supernatant was transferred to 1-cm UV cell. The difference between absorbance of the tablet containing ketamine HCl and the tablet without ketamine HCl, considered as main absorbance of ketamine HCl. To calculate the release percentages, a standard Ketamine HCl solution with 0, 40, 80, 120, 160, 200 µg in 0.1 N HCl solution was prepared and then the UV absorbance at 263 nm was measured to obtain the calibration curve (R2 = 0.9999) using UV-Vis spectrophotometer (Jasco UV-1500, Hachioji, Japan).
This text is added in line 201-215.
Comment of the reviewer:
- The diagrams in this manuscript are not standardized, such as figure 1 and figure 4. For example, FTIR requires specific labeling of the location of characteristic peaks.
Response of the authors:
Thanks for the comment. The fig.1 was standardized as recommended and replaced with the new figure. In Fig.4, the FTIR spectra were labeled to show the characteristics peaks.
Comment of the reviewer:
- The amorphous state of the Ketamine is not indicated by DSC alone, and it is recommended to supplement the XRD indication.
Response of the authors:
Thanks a lot for the valuable recommendation. That is right. To state more precisely we change lines 323-326 to:
The comparison of results obtained from DSC analysis of Ketamine and the final tablet may demonstrate change in the crystallinity of Ketamine to amorphous state during the production process. However, XRD analysis is needed to confirm the precise status.
Comment of the reviewer:
- Supplement the DSC profile of the carboxymethylcellulose hydrogel and explain that none of the shell materials had a heat absorption peak, whereas the printed tablets had an absorption peak at 150℃.
Response of the authors:
This shift can be attributed to three main factors. Firstly, during the 3D printing process, the API was placed inside the hydrogel at room temperature. As the hydrogel was injected into the four pits of the tablet, there was a slight increase in the API temperature due to heat transfer between the shell polymers and the API. The comparison of results obtained from DSC analysis of Ketamine HCl and the final tablet may demonstrate change in the crystallinity of Ketamine HCl to amorphous state during the production process. However, XRD analysis is needed to confirm the precise status. Secondly, considering that the tablet mean weight is 1,106 mg and the API weight is 165 mg, it is clear that the API is distributed within the polymers to a limited extent. Thirdly, we observed a small peak in the 150–170°C range for CMC, which had an impact on the mentioned peak shift. In summary, the final tablet exhibited a minor peak at approximately 160°C, rather than the expected 260°C, due to these three factors.
The text is edited in line 323-331.
Comment of the reviewer:
- The structure of the results and discussion sections of the article needs to be reconsidered and it is not recommended that an experiment be divided into one paragraph
Response of the authors:
The structures were revised as recommended.
Best

Reviewer 2 Report
The current research focusing the development of floating tablets of ketamine hydrochloride requires a major revision. Please address the below comments:
1. There are a couple of research articles published for floating tablets by additive manufacturing. Can authors please explain the novelty of current research when compared with the existing published works.
2. Line 14: Authors described the physical state of drug as “dissolved ketamine hydrochloride” whereas, in line 22, authors described that the drug has existed in crystalline form. Please rewrite the respective sentence to provide uniform meaning.
3. What’s the half-life of ketamine hydrochloride? Is there any marketed product for modified release formulation?
4. Since the authors are employing thermal process for developing the dosage forms, it is suggested to described the melting point, and degradation point for the drug (if any) and for polymers. In addition, please provide any information pertaining to the polymorphic changes of the drug attributing to the temperature.
5. Please maintain uniformity throughout the manuscript for “Ketamine” and “ketamine hydrochloride”.
6. Why ketamine hydrochloride is available in liquid form? Can authors please include information for freeze drying process and also purity of the procured solution.
7. Please add the trademark for Eudragit.
8. Please include information for city and state along with country for all the manufacturers (equipment and excipients).
9. Section 2.2: Can authors please clarify, if the drug is dissolved or suspended in the hydrogel? If the drug is being suspended, why the authors are concerned about the high melting point of drug? Solid form of the drug can be employed instead of employing freeze-dried drug.
10. Please provide the information for software version, and manufacturer along with city, state, and country.
11. Why did authors choose 165 mg dose for ketamine.
12. Did authors employ hot melt extrusion process? If so, please include the methodology. If not, what exactly does authors refer to “hot melt extruder”? If authors are referring HME for 3D printer, please make necessary changes. HME is a different piece of instrument which might create confusion to the readers.
13. For fabrication of tablets, did authors employ FDM 3D printing process or PAM process. Please make it clear in the methodology.
14. Please elaborate the 3D printing process. Provide additional information for printing speed.
15. Section 2.4: Please elaborate the methodologies for all the characterizations.
16. Figure 1 shows the shape of the tablets as round. Whereas figure 2 shows the shape as oval. Can authors please confirm what was the original geometry of the tablets and correct the respective Figure.
17. Section 3.4: Authors described “the absence of drug peak can be due to distribution of drug in the polymer”, do authors mean conversion of drug to amorphous state? Please clarify.
18. For all the characterizations, which formulations did authors chose. Please include it in the respective figures and results. For example, it difficult to identify which formulation the authors are referring to in Table 3.
English editing required
Author Response
Detailed Response to Reviewer # 2
Thank you for your comments and suggestion concerning our manuscript. The comments and suggestions are all valuable and very helpful for revising and improving the manuscript, as well as the important guiding significance to our researches. We have studied comments carefully and have made correction to address the reviewer’s comments.
Thank you for your comments and suggestion concerning our manuscript. The comments and suggestions are all valuable and very helpful for revising and improving the manuscript, as well as the important guiding significance to our researches. We have studied comments carefully and have made correction to address the reviewer’s comments.
Comment of the reviewer:
- There are a couple of research articles published for floating tablets by additive manufacturing. Can authors please explain the novelty of current research when compared with the existing published works.
Response of the authors:
Thank you for your comment. Based on the literature review, this study introduces two significant advancements compared to previous research. The initial innovation involves the development of oral ketamine tablets as an alternative to direct IV injection. This approach reduces hospitalization duration and mitigates the generation of the secondary metabolite 2,6-hydroxynorketamine due to pre-systemic metabolism. The second innovation is the utilization of 3D printing (additive manufacturing) technology which enables the formulation of personalized and sustained-release ketamine tablets according to the chronic or bipolar depression intensity and suicidal tendency.
To achieve the mentioned innovation, in the study, for the first time, the HME/PAS-based 3D printing method was used to applied to develop a tunable formulation as a core-shell floating system, providing tailored doses of extended-release ketamine tablets for the patients suffering from chronic depression. In this regard, a new formulation consisting Soluplus: Eudragit RS-PO in shell compartment and carboxymethylcellulose gel containing drug in core compartment was developed. We have added and revised lines 132-145 for better describing these novelties. The Core-shell word was added to the title, too.
Comment of the reviewer:
- Line 14: Authors described the physical state of drug as “dissolved ketamine hydrochloride” whereas, in line 22, authors described that the drug has existed in crystalline form. Please rewrite the respective sentence to provide uniform meaning.
Response of the authors:
Thank you for the detailed comment. “Dissolved ketamine hydrochloride” is edited to
“Ketamine HCl” in page 2 line 7 for more uniformity meaning.
Comment of the reviewer
- What’s the half-life of ketamine hydrochloride? Is there any marketed product for modified release formulation?
Response of the authors:
Ronald F Donnelly et al have reported the stability study for diluted Ketamine packaged in Glass Vials (Can J Hosp Pharm. 2013 May-Jun; 66(3): 198). The more diluted solution was stable for 12 months when stored at 4°C, 25°C, or 40°C, whereas the more concentrated solution was stable for 30 days at 25°C. The stability for diluted one decreased to 3 month upon light exposure.
Furthermore, Loyd V. Allen has reported that the formulation of Ketamine Hydrochloride 10-mg Troches is stable more than 6month.
Interestingly, the shelf life for Ketamine 50mg/ml Solution for Injection, developed by
Panpharma UK Ltd is 60 month.
Therefore, there is no commercialized ketamine tablet. But, every approved tablet must address the ICH requirements for stability. Accordingly, the extended release floating tablet of ketamine HCL have potential to show a good stability in the market.
Comment of the reviewer
- Since the authors are employing thermal process for developing the dosage forms, it is suggested to describe the melting point, and degradation point for the drug (if any) and for polymers. In addition, please provide any information pertaining to the polymorphic changes of the drug attributing to the temperature.
Response of the authors:
Thanks for the comment. Melting point and degradation point of polymers added in line 175-178. Melting point of Ketamine hydrochloride was mentioned in line 157.
Comment of the reviewer
- Please maintain uniformity throughout the manuscript for “Ketamine” and “ketamine hydrochloride”.
Response of the authors:
Thank you. It was corrected throughout of the manuscript.
Comment of the reviewer
- Why ketamine hydrochloride is available in liquid form? Can authors please include information for freeze drying process and also purity of the procured solution.
Response of the authors:
Thanks for the precise attention. Actually, we had problems in supplying the raw material of Ketamine HCl in powder form. So, the drug in solution form was purchased and freeze dried. The determination of Ketamine hydrochloride content in the lyophilized powder showed a purity of about 98.7%.
To fabricate 3D printed tablet by hot melt extrusion, melting API with polymers is a common method. However, in the present study it was not possible to melt API and polymers simultaneously because of the higher melting point of Ketamine HCl powder (~262°C), compared to the maximum melting capability of the 3D printing device (~210°C). Furthermore, thermal degradation of Ketamine HCl during hot melt extrusion was another our concern. To address the problems, it was tried to formulate core-shell tablet of ketamine. Ketamine HCl was solubilized within a hydrogel, and this hydrogel was subsequently deposited by second nozzle (Pressure assisted syringe without thermal treatment) into the shell during the printing process. The dosage form is not strange and it is similar to Colpermin and gelatin capsules, but in an extended release form.
Comment of the reviewer
- Please add the trademark for Eudragit.
Response of the authors:
Thanks a lot. It was corrected throughout of the manuscript.
Comment of the reviewer
- Please include information for city and state along with country for all the manufacturers (equipment and excipients).
Response of the authors:
The information was included as recommended.
Comment of the reviewer
- Section 2.2: Can authors please clarify, if the drug is dissolved or suspended in the hydrogel? If the drug is being suspended, why the authors are concerned about the high melting point of drug? Solid form of the drug can be employed instead of employing freeze-dried drug.
Thanks for the precise comment. To remove any concern of thermal degradation or decomposition of Ketamine HCl over time in hot melt container of 3D printer, before extrusion and printing, we decided to dissolve the drug in hydrogel. Additionally, Ketamine HCl in mixture with polymers produced a non-extrudable hard filament after melt extrusion.
Comment of the reviewer
- Please provide the information for software version, and manufacturer along with city, state, and country.
Response of the authors:
The information was added as requested in lines 171-172.
Comment of the reviewer
- Why did authors choose 165 mg dose for ketamine.
Response of the authors:
Thanks a lot for the valuable comments. We corrected our mistake, it was 162.5 mg. Because only 20%-25% of orally administrated ketamine reaches systemic circulation, and doses of about 2-2.5mg/kg may be needed to be administrated to achieve equivalence to intravenously administrated ketamine, 162.5 mg pure ketamine hydrochloride was dose amount, considering the mean ideal body weight of 65kg for a healthy human.
Comment of the reviewer
- Did authors employ hot melt extrusion process? If so, please include the methodology. If not, what exactly does authors refer to “hot melt extruder”? If authors are referring HME for 3D printer, please make necessary changes. HME is a different piece of instrument which might create confusion to the readers.
Response of the authors:
Thanks for the great comment. Actually, in the study for fabrication of the extended release tablet of Ketamine using 3D printing, a core-shell Tablet was developed so that the shell compartment was printed thermoplastic polymers using HME nozzle and the core compartment was printed, CMC hydrogel containing drug, using pressure-assisted-syringe (PAS) as second nozzle.
The methodology text was edited in abstract and in lines 183-187 to clarify the process.
Comment of the reviewer
- For fabrication of tablets, did authors employ FDM 3D printing process or PAM process. Please make it clear in the methodology.
Response of the authors:
We made it clear in the methodology as mentioned in response to q12.
Comment of the reviewer
- Please elaborate the 3D printing process. Provide additional information for printing speed.
Response of the authors:
Thanks for the great comment. We add the speed of printing for both nozzle in the text in line 181 and 187:
“The shell of the tablets was firstly printed under the temperature of 150°C –180°C, depending on the specific formulation (F1 at 180°C, F2 at 170-180°C, F3 at 160-170°C and F4 at 150-160°C) at a speed of 18mm/sec. After printing 20 layers by the HME nozzle, the second extruder, PAM, injected Ketamine HCl-CMC gel inside the four pits of the tablet under ambient temperature (25°C) at speed of 20mm/sec”
Comment of the reviewer
- Section 2.4: Please elaborate the methodologies for all the characterizations.
Response of the authors:
It was corrected as recommended.
Comment of the reviewer
- Figure 1 shows the shape of the tablets as round. Whereas figure 2 shows the shape as oval. Can authors please confirm what was the original geometry of the tablets and correct the respective Figure.
Response of the authors:
The resolution and the image sizes are modified to show the oval shape tablet. The images was labelled by ruler to show dimensions.
Comment of the reviewer
- Section 3.4: Authors described “the absence of drug peak can be due to distribution of drug in the polymer”, do authors mean conversion of drug to amorphous state? Please clarify.
Response of the authors:
Thanks a lot. That is right. Accordingly we change the line 318-331 as follow:
Notably, we observed an endothermic peak at 160°C in the DSC thermogram of the final tablet, which differed from the 260°C peak observed in the thermogram of pure Ketmine HCl. This shift can be attributed to three main factors. Firstly, during the 3D printing process, the API was placed inside the hydrogel at room temperature. As the hydrogel was injected into the four pits of the tablet, there was a slight increase in the API temperature due to heat transfer between the shell polymers and the API. The comparison of results obtained from DSC analysis of Ketamine HCl and the final tablet may demonstrate change in the crystallinity of Ketamine HCl to amorphous state during the production process. However, XRD analysis is needed to confirm the precise status. Secondly, considering that the tablet mean weight is 1,106 mg and the API weight is 165 mg, it is clear that the API is distributed within the polymers to a limited extent. Thirdly, we observed a small peak in the 150–170°C range for CMC, which had an impact on the mentioned peak shift. In summary, the final tablet exhibited a minor peak at approximately 160°C, rather than the expected 260°C, due to these three factors.
Comment of the reviewer
- For all the characterizations, which formulations did authors chose? Please include it in the respective figures and results. For example, it difficult to identify which formulation the authors are referring to in Table 3.
Response of the authors:
Thanks a lot for the comment. Please refer to lines 247-250: “Among the four formulations, F4 with 80/20 (W/W) ratio of Soluplus®/Eudragit® RS-PO revealed the best performance in terms of extrudability, printability, floating capability, and entire release within 12 h. Therefore, this formulation was selected for further characterization.”
So, table 3 shows content uniformity results obtained from F4 formulation. We specify in Table 3 caption.
Best

Reviewer 3 Report
The manuscript title “Fabrication of a Controlled-Release Floating Tablet of Ketamine Using 3D Printing Technique for Management of Refractory Depression and Chronic Pain” is an interesting on the 3D printing techniques for management of Chronic Pain and Refractory Depression. This manuscript is acceptable for publication with few suggestions.
Suggestions/Comments:
1. Correct the temperature range for tablet shell extrusion e.g., in abstract it is mentioned as 150-160 °C while in material and methods section it is mentioned as 150-180 °C.
2. Why the batch F4 exhibiting the best extrudability and floatability results author should discuss in the discussion part with some supporting literature.
3. Since Ketamine HCl is highly polar molecule, author should write and discuss for the permeability through gastrointestinal membrane?
4. The % transmittance in FTIR Figure 4. Is not uniform?
5. The missing of stability testing of the tables author should add at least Initial, 1, 2 and 3 months.
6. Why author have selected Soluplus and Eudragit RS-PO as shell former? and should compared with any other polymers?
7. Why have you selected tablet shell design with 4 pits? And have you compared it with other designs with better floatability and sustained release profile?
8. Can compare buoyancy test by incorporating body temperature condition (37 ± 0.5 °C) and RPM.
9. Figure 7. Author shows the floating of the tablets after 12 h using 0.1 N HCL solution, is really tablets still remained float after 12 hr, Although Figure 6 demonstrating the100% release of ketamine in 12 hr. How the tablet would remain it’s in original nature Please justify?
10. Author should also show the picture of 0, 2, 6, and 12 hr time interval during floating and release.
11. Author should mention the floating lag time and total floating time of the 3D printing tablets.
12. Describe which release kinetic model fits based on drug release profile.
Minor language and grammar editing is required.
Author Response
Detailed Response to Reviewer # 3
Thank you for your comments and suggestion concerning our manuscript. The comments and suggestions are all valuable and very helpful for revising and improving the manuscript, as well as the important guiding significance to our researches. We have studied comments carefully and have made correction to address the reviewer’s comments.
Comment of the reviewer
- Correct the temperature range for tablet shell extrusion e.g., in abstract it is mentioned as 150-160 °C while in material and methods section it is mentioned as 150-180 °C.
Response of the authors:
Thanks for the comment. 150 -180°C in material methods refers to 4 formulations but in abstract it refers to F4 only. F1 at 180°C, F2 at 170-180°C, F3 at 160-170°C and F4 at 150-160°C. Related information is added in line 145 page 8.
Comment of the reviewer
- Why the batch F4 exhibiting the best extrudability and floatability results author should discuss in the discussion part with some supporting literature
- Why author have selected Soluplus and Eudragit RS-PO as shell former? and should compared with any other polymers?
Question 2 and 6 has same answer.
Response of the authors:
As it is mentioned in table 2. All formulations had good floatability and extrudability but the release trends were different. Soluplus® as a graft copolymer constructed from PCL, PVA, and PEG components, with amphiphilic characteristics is extrudabledue to its low glass transition temperature (Tg) [1]. Eudragit RS PO as a low porosity and low permeability polymer has been used for making controlled release tablets in a pH independent release manner. Furthermore, the has been used in different studies for making floating tablets [2].
The above text was added in section 3.1. Tablet formulation and 3D printing process.
Comment of the reviewer
- Since Ketamine HCl is highly polar molecule, author should write and discuss for the permeability through gastrointestinal membrane?
Response of the authors:
Thanks a lot. We wrote about ketamine permeability in Lines 251-253.
“Ketamine, a BCS Class 1 drug, has high passive permeability with no P-glycoprotein mediated efflux so there will be no rate limitation for intestinal transmucosal absorption.”
Comment of the reviewer
- The % transmittance in FTIR Figure 4. Is not uniform?
Response of the authors:
The output data as FTIR spectrum results was obtained from an instrument in a reliable company and the % transmittance was set based on the intensity of absorption. Although the device is not available, the appearance of the peaks has the necessary resolution for this qualitative analysis.
Comment of the reviewer
- The missing of stability testing of the tables author should add at least Initial, 1, 2 and 3 months.
Response of the authors:
The stability testing of the tablet according to ICH guideline is presented as follow:
|
Stability Specification |
||||||
|
Row |
Test item |
Time (Month) |
0 |
1 |
3 |
6 |
|
Standard |
|
|
|
|
||
|
1 |
Appearance |
White Oval Tablet |
Complies |
Complies |
Complies |
Complies |
|
2 |
Hardness |
Above 50 Kg |
Complies |
Complies |
Complies |
Complies |
|
3 |
Friability |
Less Than 1% |
Complies |
Complies |
Complies |
Complies |
|
4 |
Assay |
90.0–110.0% |
100.1 |
99.8 |
99.4 |
99. 1 |
Comment of the reviewer
- Why have you selected tablet shell design with 4 pits? And have you compared it with other designs with better floatability and sustained release profile?
Response of the authors:
Ketamine HCl melting point is high (262°C), as a result we decided to put it into hydrogel and inject inside the tablet. At first, we designed a round tablet but it did not have enough space for hydrogel and API. When we consider an increased diameter in the design, a massive tablet which was difficult to swallow came out. Therefore, we changed the design shape to oval which was easier to swallow.
Regarding 165 mg of Ketamine HCl, the best design is 4 pits which is giving us the most vacant space inside the tablet in order to inject hydrogel containing ketamine HCl into the four pits. Moreover, designing pits decreases the shell weight and density, eventually helping the tablet to remain float.
Comment of the reviewer
- Can compare buoyancy test by incorporating body temperature condition (37 ± 0.5 °C) and RPM.
- Figure 7. Author shows the floating of the tablets after 12 h using 0.1 N HCL solution, is really tablets still remained float after 12 hr, Although Figure 6 demonstrating the100% release of ketamine in 12 hr. How the tablet would remain it’s in original nature please justify?
- Author should also show the picture of 0, 2, 6, and 12 hr time interval during floating and release.
Response of the authors to the questions 8 – 10:
Thanks for these comments, because we deleted some pics from our article, a mistake was happened. The pic is for 1 hour of the test. Buoyancy test in 37°C and RPM 100 was performed and it is added in page 12 in the revised version of the manuscript. Apparantly, the tablet kept floating but in the swelling form.
Figure 7. Demonstration of 3D-printed tablet in HCl solution of 0.1 N at 37°C after a) after 1 h; b) 3h; c) 9 h and d) 12 h.
- Author should mention the floating lag time and total floating time of the 3D printing tablets.
The tablet was float from first to end of 12 hours.
- Describe which release kinetic model fits based on drug release profile.
The visual observations of the dissolution test confirmed the swelling mechanism, which is one of the characteristics of Eudragit RS. Our previous experience in a study showed diffusion is another mechanism.
Actually, various release kinetic models including first-order, Higuchi, Hixson Crowell cube root, Korsemeyer–Peppas, and zero-order model can be exploited to fit the in vitro drug release data, which we have not perform.
Best

Round 2
Reviewer 1 Report
The main body of this manuscript describes a method for printing ketamine controlled-release oral tablets utilizing hot melt extrusion (HME), and this manuscript is substantially in accordance with the requirements for publication
Reviewer 2 Report
All the comments are well addressed, the revised version can be accepted for publication.
